# Association of Breastfeeding and Early Childhood Caries: A Systematic Review and Meta-Analysis

**DOI:** 10.3390/nu16091355

**Published:** 2024-04-30

**Authors:** Sheetal Kiran Shrestha, Amit Arora, Narendar Manohar, Kanchana Ekanayake, Jann Foster

**Affiliations:** 1School of Health Sciences, Western Sydney University, Locked Bag 1797, Penrith, NSW 2751, Australia; 2Health Equity Laboratory, Campbelltown, NSW 2560, Australia; 3Discipline of Child and Adolescent Health, The Children’s Hospital at Westmead Clinical School, Faculty of Medicine and Health, The University of Sydney, Westmead, NSW 2145, Australia; 4Translational Health Research Institute, Western Sydney University, Campbelltown, NSW 2560, Australia; 5Oral Health Services, Sydney Local Health District and Sydney Dental Hospital, NSW Health, Surry Hills, NSW 2010, Australia; 6Blackdog Institute, Hospital Road, Randwick, NSW 2031, Australia; 7University of Sydney Library, The University of Sydney, Camperdown, NSW 2006, Australia; 8School of Nursing and Midwifery, Western Sydney University, Locked Bag 1797, Penrith, NSW 2751, Australia; 9School of Nursing and Midwifery, University of Canberra, Bruce, ACT 2617, Australia; 10Ingham Research Institute, Liverpool, NSW 2170, Australia

**Keywords:** breastfeeding, early childhood caries, preschool children, oral health, dental caries

## Abstract

Early childhood caries (ECC) is a growing public health concern worldwide. Although numerous systematic reviews have been published regarding the association between breastfeeding and early childhood caries (ECC), the results remain inconclusive and equivocal. This systematic review synthesises the evidence on the association between breastfeeding and ECC. Five electronic databases and backward citation chasing were performed from inception until May 2023. A total of 31 studies (22 cohort studies and 9 case-control studies) were included in this review. The meta-analysis of the case-control studies showed statistically significant fewer dental caries in children who were breastfed for < 6 months compared to those who were breastfed for ≥6 months (OR = 0.53, 95% CI 0.41–0.67, *p* < 0.001). There was a statistically significant difference in dental caries between children who were breastfed for <12 months and those who were breastfed for ≥12 months (RR = 0.65, 95% CI 0.50–0.86, *p* < 0.002). Similarly, there was a statistically significant difference in dental caries in children who were breastfed for < 18 months compared to those who were breastfed for ≥18 months (RR = 0.41, 95% CI 0.18–0.92, *p* = 0.030). Nocturnal breastfeeding increases the risk of ECC compared with no nocturnal breastfeeding (RR = 2.35, 95% CI 1.42–3.89, *p* < 0.001). The findings suggest breastfeeding for more than 12 months and nocturnal breastfeeding increase the risk of ECC.

## 1. Introduction

Oral health is the state of well-being of the mouth and its associated structures that enable individuals to perform their essential functions without any pain or discomfort, as well as maintain their psychosocial health [1]. About 3.5 billion people in the world suffer from oral diseases, including dental caries [1]. Early childhood caries (ECC) is the most common oral disease among children [2]. The American Academy of Paediatric Dentistry (AAPD) defines ECC “as the presence of one or more decayed (non-cavitated or cavitated lesions), missing (due to caries), or filled tooth surfaces in any primary tooth in a child 71 months of age or younger” [3]. Globally, 514 million children, constituting 43% of the paediatric population, suffer from dental caries in their primary teeth [1]. Dental caries among young children is a widespread issue, with 21% of American children aged two to five years experiencing dental caries [4]. Similarly, a quarter of English children exhibit dental caries before starting school [5]. Likewise, the Australian National Child Oral Health Study reported that 34.3% of children between the ages of five and six experience dental caries in their primary teeth, with 26.1% of those cases left untreated [6]. The plight of ECC in the developing world is worse. For example, in India, approximately one in two children suffers from ECC [7].

Early childhood caries develops along the gingival (gum) margin as a white spot lesion [8] and may progress quickly, resulting in health adversities such as toothache and infection [5]. Furthermore, in severe cases of ECC, surgery and/or hospital admission may be required for treatment, thus affecting not only the child but also the family [9].

ECC has a multifactorial aetiology [10]. ECC is strongly influenced by socioeconomic status (SES) and is more prevalent in children from families with low SES and in developing countries [7]. Other factors, such as a mother’s education, attitude towards oral health, the presence of enamel defects, high levels of mutans streptococci, and feeding habits, increase the risk of ECC [11]. Feeding habits such as breastfeeding [12] and the age at which free sugars, including those naturally occurring in fruit juices or honey, are introduced, as well as the frequency of their consumption, play a crucial role in the development of ECC [13].

The association between breastfeeding and ECC is one of the most debated risk factors for ECC. Several studies have indicated that prolonged and exclusive breastfeeding does not contribute to the development of ECC in preschool children [14,15,16,17]. However, several other studies have reported that breastfeeding beyond 12 months and nocturnal breastfeeding increase the risk of ECC. This conundrum has kept the association between breastfeeding and oral health debatable and inconclusive. As a result, medical and dental organisations provide different recommendations, as outlined below. The World Health Organisation (WHO) recommends that all infants be exclusively breastfed for 6 months and complementary breastfeeding continue for 2 years or more [18]. In Australia, the National Health and Medical Research Council (NHMRC) recommends exclusive breastfeeding until around 6 months of age and continuing complementary breastfeeding until 12 months of age and beyond, provided that both the mother and child wish to do so [19]. The American Association of Paediatric Dentistry (AAPD) advocates breastfeeding for a duration of 12 months to promote optimum health and developmental outcomes for the infant and establish psychosocial wellbeing in infants [20]. However, the AAPD cautions against unrestricted nocturnal breastfeeding following the eruption of the first primary tooth, arguing that nocturnal feeding can increase the risk of ECC. Likewise, some studies suggest that children who are breastfed may exhibit a higher likelihood of developing dental caries than children who do not breastfeed [12,21].

Several systematic reviews [22,23,24,25,26,27,28,29] have explored the association of breastfeeding and ECC (Appendix A). However, the evidence was assessed to be inconclusive and equivocal. Moreover, most of the above reviews were deemed to be of low quality, except for one [26], which was of moderate quality (Appendix A). Moynihan et al. (2019) [26] did not conduct any meta-analysis, nor was it solely focused on breastfeeding and ECC. Most of the reviews included observational studies, including cross-sectional studies, which do not establish any causal evidence [22,23,24,25,26,28,29]. This systematic review emphasises scientific rigour by incorporating only longitudinal observational studies, specifically cohort and case-control studies, which are considered to possess a higher level of evidence among the observational studies [30]. Therefore, this review is designed to assess and consolidate the available evidence from cohort and case-control studies to provide a high-quality, comprehensive synthesis on the effect of breastfeeding on ECC among preschool children. Additionally, this review aims to investigate the prevalence of dental caries (ECC) in preschool children who were breastfed for different durations (<4 months, <6 months, <12 months, <18 months, <24 months, ≥24 months) and those who received nocturnal breastfeeding.

Hence, this review and meta-analyses address the following research questions:Is there any association between breastfeeding and ECC?Is the prevalence of ECC different among children breastfed for different durations?Does nocturnal breastfeeding increase the risk of ECC?

## 2. Methods

This systematic review followed the JBI methodology for systematic reviews of aetiology and risk [31], and the results were presented as per the PRISMA 2020 guidelines [32]. The protocol of this systematic review has been registered in the PROSPERO International Prospective Register of Systematic Reviews (CRD42023442205).

### 2.1. Eligibility Criteria

The studies included in the review were of cohort (prospective and retrospective) and case-control design. Randomised controlled trials (RCTs), quasi-experimental studies, cross-sectional studies, and single case reports were excluded. Furthermore, grey literature, systematic reviews, literature reviews, umbrella reviews, and scoping reviews were also excluded from this review. The eligibility criteria are outlined in detail in Table 1.

### 2.2. Information Sources

The search was conducted in five databases without any date restrictions: MEDLINE (Ovid), Scopus, Web of Science (ISI), Embase (Ovid), and CINAHL (EBSCO). The search was conducted until 13 May 2023. The reference lists of the included studies and relevant past published systematic reviews were searched manually to ensure the inclusion of all relevant studies.

### 2.3. Search Strategy

The reviewers (SS, JF, and AA) devised the research question and search terms based on the P(I/E)CO (Population Intervention/Exposure Comparator Outcome) criteria. A logic grid was drafted using medical subject headings (MeSH) terms and keywords in collaboration with a health sciences librarian (KE). A preliminary search was conducted on MEDLINE (Ovid). The search strategy was adapted and modified for the remaining databases (Appendix A).

### 2.4. Study Selection

Studies were identified through searching the five databases and were imported into Covidence. Two reviewers (SS and JF) independently screened the titles and abstracts of the imported articles against the inclusion criteria. Additional information from study authors was sought for studies with uncertain eligibility. In cases where there were no responses from the study authors after three attempts, the study was screened based on the available information. Any article with ambiguous eligibility was discussed between the two reviewers to reach a conclusion. In any instance where the two reviewers could not come to an agreement, a third reviewer (AA) was involved to seek a resolution. Duplicates identified in Covidence were removed. The reference lists of studies included in the full-text review were also screened for additional studies. A list of the excluded studies with reasons for their exclusion is provided in Appendix A. The study selection process followed the PRISMA checklist, and the flow diagram is presented as Figure 1.

### 2.5. Data Collection Process and Data Items

The data were extracted using JBI SUMARI data extraction tools. Two reviewers (SS and JF) calibrated the information collected from two studies to ensure consistency in data extraction. The same two reviewers then extracted the data independently from the selected studies (Appendix A). The information extracted from the included studies was the name of the authors and publication year, study design, setting and sample size, country, demography of the participants, breastfeeding data (method, duration, and timing), follow-up time, dental caries measures (dmft index, dmfs, index, dfs, ICDAS), study results, funding, and conclusion. A maximum of 3 attempts were made to contact the authors of potentially eligible studies for any unclear or missing information.

### 2.6. Data Synthesis

Cohort studies and case-control studies were included in this review. Effect sizes were expressed as risk ratios, odds ratios, or prevalence ratios for dichotomous data, and their 95% confidence intervals (CI) were calculated for analysis. Cohort estimates were presented as risk ratios with a 95% CI, and case-control estimates were presented as odds ratios with a 95% CI. Meta-analyses were conducted comparing children who were breastfed and those who were not; children breastfed for different durations (<4 months vs. ≥4 months; <6 months vs. ≥6 months; <12 months vs. ≥12 months; <18 months vs. ≥18 months; <24 months vs. ≥24 months); and those who received nocturnal breastfeeding vs. those who did not.

The meta-analyses were conducted using the statistical software “Review Manager” (RevMan Web Version 7.9.0). A fixed-effect model for meta-analysis was used in the first instance to combine the data where possible. When the heterogeneity was substantial, a random effects model was used. Heterogeneity was evaluated using the Chi^2^ test and I^2^ analysis. For the Chi^2^ test, a *p* value of less than 0.10 was considered for heterogeneity. The I^2^ value was considered according to the *Cochrane Handbook for Systematic Reviews of Interventions* [33] and the *JBI Manual for Evidence Synthesis* [34], where 0% to 40% was considered not important, 30% to 60% represented moderate heterogeneity, 50% to 90% represented substantial heterogeneity, and 75% to 100% represented considerable heterogeneity [33]. Studies that were inappropriate to be included in the meta-analysis due to their study design or reporting of the outcome measures were analysed, interpreted individually, and reported narratively.

### 2.7. Quality Assessment and Risk of Bias

Each study included in this review was independently assessed for their methodological quality by two reviewers (SS and JF) utilising the Joanna Briggs Institute critical appraisal tool for cohort studies (retrospective and prospective) and case-control studies [31]. The results of the JBI quality assessments are reported in Appendix A. All the studies were included in the review, regardless of their methodological quality. The third reviewer (AA) was consulted to seek a resolution in case of any disagreements.

## 3. Results

A total of 2527 studies were identified from the initial search. After 640 duplicates were removed, 1887 studies were screened. After removing 1793 records post title and abstract screening, a total of 118 studies were screened for full-text eligibility. After the exclusion of 88 studies (Appendix A), 31 studies matching the inclusion criteria were included in this review. A total of 26 studies were included in the meta-analysis, and five were reported narratively (Figure 1). All the studies were in English and were published between 1994 and 2023. A total of 15,236 pre-school children were analysed.

### 3.1. Study Characteristics

Out of the 31 studies, 22 were cohort studies [35,36,37,38,39,40,41,42,43,44,45,46,47,48,49,50,51,52,53,54,55,56], and 9 were case-control studies [57,58,59,60,61,62,63,64,65]. Among the cohort studies, six were nested studies, one was nested within a cross-sectional study [36], and the remaining five were nested in randomised controlled trials [38,40,41,42,51]. Four of the cohort studies were retrospective in design [43,45,49,53]. Due to the high heterogeneity among the studies, most of the results were presented using random effects.

Eleven studies carried out dental examinations in clinical settings of dental clinics, hospitals, or universities [35,36,43,44,45,46,47,53,57,60,62]. Fourteen studies carried out dental examinations in field settings [37,38,40,42,48,49,50,51,54,55,58,59,61,64]. Four studies were unclear about their setting for dental examinations [39,56,63,65].

### 3.2. Study Settings

Seventeen studies [35,36,38,40,41,43,48,49,50,51,54,58,59,60,61,62,63] were conducted in middle-income countries [66] whereas fourteen [37,39,42,44,45,46,47,52,53,55,56,57,64,65] were carried out in high-income countries.

Most of the studies were conducted in Brazil [35,36,38,40,41,43,49,51]. Five studies were conducted in Japan [47,52,53,55,56]. Four studies were conducted in Australia [39,42,46,64]. Two studies were conducted in India [58,59], and two were conducted in Thailand [48,50]). Among the high-income countries, one study was conducted in the USA [44], one was conducted in Canada [65], one was undertaken in Italy [45], one in Scotland [37], and one in the Czech Republic [57]. Refs. [61,63] were undertaken in Tanzania and South Africa, respectively, while [54,62] were conducted in China and Myanmar, respectively.

### 3.3. Early Childhood Caries

The main outcome, ECC, was reported and classified differently amongst the studies. The outcomes for ECC were reported as dmfs (decayed, missing, and filled surfaces), dmft (decayed, missing, and filled teeth), dft (decayed, filled teeth), and dfs (decayed, filled surface). S-ECC (severe early childhood caries) was also reported and refers to one or more decayed, missing (due to caries), or filled tooth surfaces in the primary (baby) maxillary (upper) and anterior (front) teeth.

Eight studies [37,39,43,48,58,60,61,65] reported dental caries as dmfs ECC. Five studies [38,46,50,51,63] reported dental caries as dmfs S-ECC. Seven studies [35,42,47,49,52,55,64] reported dmft ECC. Two studies [53,56] reported ECC as the presence or absence of dental caries (cavitated and non-cavitated lesions). Two studies [40,41] reported ECC and S-ECC as modified dmft and dmfs. One study [44] reported dfs ECC. One study [54] only reported ECC as a frank cavitation. Barosso et al. (2021), Ganesh et al. (2022), and Majorana et al. (2014) [36,45,59] were the only three studies reporting ECC based on the International Caries Detection Assessment System (ICDAS) criteria.

### 3.4. Methodological Quality and Risk of Bias Assessment

The quality assessments of the 21 cohort studies and 9 case-control studies were conducted using appropriate JBI tools for cohort studies and case-control studies, respectively [31]. The critical appraisals for the studies are presented in Appendix A. The critical appraisal was conducted by two reviewers separately (SS and JF). Any differences were resolved through discussion until agreement was reached.

### 3.5. Cohort Studies

Chaffee et al. (2014), Devenish et al. (2020), Manohar et al. (2021), Nirunsittirat et al. (2016), Peltzer et al. (2015), and Yokoi et al. (2020) [38,39,46,48,50,55] were the only studies that fulfilled all the criteria among the cohort studies (Appendix A). The similarity between the two groups (Q1), the similarity (Q2), and the validity (Q3) of the exposure were addressed by all the studies except Yonezu et al. (2006) [56], which was unclear for Q1 and Q3.

Confounding factors (Q4) were addressed by all the studies except Majorana et al. (2014), where it was unclear, and Yonezu et al. (2006) did not address the criteria. All studies mentioned the strategies to address confounders (Q5) except for Abanto et al. (2023), Feldens et al. (2010), and Majorana et al. (2014) [35,40,45], which were unclear, and Yonezu et al. (2006) [56] did not address the criteria. A third of the cohort studies [36,37,47,49,53,54,56] were either unclear or failed to mention if the participants were free of the outcome at the start of the study (Q6); the participants in the remaining two-thirds of the studies were free of the outcome at the start of the study.

Except for Yonezu et al. (2006) [56], the outcomes in all the studies were measured in a valid way (Q7). The follow-up time was adequate in all the studies (Q8). All but five studies [36,37,44,50,56] completed follow-up or addressed the loss of follow-up (Q9). All studies except Bernabé et al. (2017), Hong et al. (2014), and Peres et al. (2017) [37,44,51] addressed Q10 (strategies to deal with loss to follow up), while Majorana et al. (2014), Tanaka et al. (2013), and Yonezu et al. (2006) [45,52,56] were rated unclear. All the studies utilised appropriate statistical analysis (Q11). Overall, the studies were of moderate to high quality (Appendix A).

### 3.6. Case-Control Studies

For the case-control studies, Ganesh et al. (2022) [59] and Lima et al. (2016) [60] were the only studies to fulfil all the criteria. All studies except Matee et al. (1994) [61] and Seow et al. (2009) [64] fulfilled Q1 (participants were comparable other than the presence of disease). The cases and controls were appropriately matched (Q2) in all the studies except in Matee et al. (1994) [61], whereby it was unclear. All the studies [58,59,60,61,62,63,64,65] addressed Q3, except for Cvanova et al. (2022) [57], and all the studies [57,59,60,61,64,65], except Dabawala et al. (2017) [58], Qin et al. (2008) [62], and Roberts et al., (1994) [63], addressed Q4.

All the studies addressed Q5–Q10, except Matee et al. (1994) [61]. It did not mention any strategies to deal with confounders (Q7). Overall, the majority of the studies were of moderate to high quality (Appendix A).

### 3.7. Effects of Breastfeeding on Early Childhood Caries

#### 3.7.1. Children Who Were Breastfed versus Children Who Were Not Breastfed

Fifteen studies [37,39,41,42,45,46,49,50,53,56,58,62,63,64,65] compared dental caries in children who were breastfed and children who were not breastfed and were included in the meta-analysis. Two separate analyses were conducted based on the study design. Of these fifteen studies, ten were of cohort design [37,39,41,42,45,46,49,50,53,56], and five were case-control studies [58,62,63,64,65].

In terms of cohort studies, the pooled estimates (Figure 2) show no statistically significant difference for dental caries between breastfed and non-breastfed children (RR 0.98, 95% CI 0.77 to 1.26; participants = 8506; studies = 10; I^2^ = 91%, *p* = 0.89). To investigate the sensitivity, five studies [37,41,45,53,56] were removed. The heterogeneity was reduced to 0% (*p* = 0.048), and the pooled estimate suggested breastfeeding may be protective against ECC compared to no breastfeeding (RR = 0.82, 95% CI 0.71 to 0.95) (not shown in the figure). However, the result should be interpreted with caution due to the high heterogeneity among the studies.

In terms of case-control studies, the pooled estimates (Figure 3) show no significant difference in the dental caries experience between breastfed and non-breastfed children (OR 1.04, 95% CI 0.78 to 1.38; participants = 1022; studies = 5; I^2^ = 14%, *p* = 0.81).

#### 3.7.2. Children Who Were Breastfed at Night versus Children Who Were Not

Seven studies reported nocturnal breastfeeding [39,47,50,54,59,60,61], and only three cohort studies [39,47,54] were able to be included in the meta-analysis. The pooled estimate for these cohort studies showed nocturnal breastfeeding could increase the risk of ECC compared with no nocturnal breastfeeding (RR 2.35, 95% CI 1.42 to 3.89; participants = 1590; studies = 3; I^2^ = 86%, *p* = 0.0008). After excluding the study with the possible risk of bias [39], the heterogeneity was found to be not important (I^2^ = 33%, *p*= 0.22) and the pooled estimate showed higher risk for the nocturnal breastfeeding group (RR = 3.05, 95% CI 2.40 to 3.87, *p* < 0.0001) The I^2^ indicates substantial heterogeneity and therefore the result should be interpreted with caution. See Figure 4.

The following four studies were not able to be included in the meta-analysis due to the study design or method of reporting the outcome, and their results were provided narratively. Ganesh et al. (2022) [59] was the only study dedicated to sleep-time feeding (both nighttime and daytime) practices. The study reported on ECC in children with various modes of feeding (breast, bottle, or other) during the beginning of sleep, early morning hours of sleep, and course of sleep. The aOR for breastfeeding at the beginning of sleep, during sleep, and in the early morning hours of sleep were found to be 6.7, 6.5, and 3.7, respectively (*p* = 0.001). The study found a strong association between sleep-time feeding and ECC. Lima et al. (2016) [60] found that children who were no longer breastfeeding during the night at 16 months were less likely to be diagnosed with ECC compared to children who engaged in nocturnal breastfeeding at 16 months (OR 0.51 (CI = 0.39–0.65), *p* < 0.001). Similarly, Matee et al. (1994) [61] found a strong association between ECC and night feeding habits with the breast nipple in the mouth of the sleeping infant (0 versus 5, OR = 17.8 (6.3–50.3), *p* = 0.0001). Peltzer and Mongkolchati (2015) [50] found nocturnal breastfeeding at 12 months (suckle to sleep when going to bed) to be associated with S-ECC (251/563 children). No other data were provided.

### 3.8. Association between Dental Caries and Breastfeeding Duration

#### 3.8.1. Duration of Breastfeeding: <4 Months and ≥4 Months

Two studies [46,50] compared dental caries in children who were breastfed < 4 months and ≥4 months (Figure 5). No significant difference was found in the analysis (RR 0.95, 95% CI 0.80 to 1.13; participants = 1017; studies = 2; I^2^ = 26%, *p* = 0.58).

#### 3.8.2. Duration of Breastfeeding: <6 Months and ≥6 Months

Nine studies [37,39,44,46,48,52,57,60,62] compared dental caries in children who were breastfed < 6 months and ≥ 6 months (Figure 6 and Figure 7). Of these, six studies were of cohort design [37,39,44,46,48,52] and were included in the meta-analysis. No significant difference was found in the analysis (RR 0.97, 95% CI 0.88 to 1.06; participants = 4850; studies = 6; I^2^ = 0%, *p* = 0.48).

However, meta-analysis of two case-control studies [60,62] showed statistically significant fewer dental caries in the <6 months group compared to the ≥6 months group (OR 0.53, 95% CI 0.41 to 0.67; participants = 1201; studies = 2; I^2^ = 0%, *p* < 0.00001).

The results of the Cvanova et al. (2022) [57] study are presented narratively due to the method of reporting the outcome. The authors found statistically significant more cases of dental caries in children breastfed for ≤6 months compared to children breastfed > 6 months [OR (multivariate) = 2.71; 95% CI 1.45 to 5.07].

#### 3.8.3. Duration of Breastfeeding: <12 Months and ≥12 Months

Eight studies comparing the duration of breastfeeding for <12 months and ≥12 months were able to be included in the meta-analysis [35,38,39,40,42,46,48,52]. All the studies were of cohort design and showed a statistically significant difference in dental caries between the <12 months group and the ≥12 months group (RR 0.65, 95% CI 0.50 to 0.86; participants = 4365; studies = 8; I^2^ = 90%, *p* < 0.002). The I^2^ indicates substantial heterogeneity, and therefore the result should be interpreted with caution. See Figure 8. Three studies [35,42,48] were removed during sensitivity analysis, where the heterogeneity dropped to 0% (*p* = 0.61). The result was still statistically significant (RR = 0.70, 95% CI 0.62 to 0.80, *p* < 0.0001).

#### 3.8.4. Duration of Breastfeeding: <18 Months and ≥18 Months

Six cohort studies [47,48,52,53,55,56] reported breastfeeding for <18 months and breastfeeding ≥ 18 months. Five cohort studies comparing the duration of breastfeeding for <18 months and ≥18 months were able to be included in the meta-analysis [47,48,52,53,56]. There were statistically significant fewer dental caries in the <18 months group compared to the ≥ 18 months group (RR 0.41, 95% CI 0.18 to 0.92; participants = 2332; studies = 5; I^2^ = 93%, *p* = 0.03) (Figure 9). After removing two studies [48,52] during sensitivity analysis, there was no heterogeneity among the studies (I^2^ = 0%, *p* = 0.45) and the pooled estimate remained statistically significant, favouring breastfeeding for <18 months (RR = 0.26, 95% CI 0.16 to 0.42). 

Due to the method of reporting of the outcomes, the result of Yokoi et al. (2020) [55] is reported narratively. They observed prolonged breastfeeding significantly increases risk of ECC [OR = 1.71; 95% CI (1.15–2.55), *p* < 0.001].

#### 3.8.5. Duration of Breastfeeding: <24 Months and ≥24 Months

Four studies comparing the duration of breastfeeding (<24 months and ≥24 months) were able to be included in the meta-analysis [35,36,42,51]. There was no statistically significant difference in dental caries in the <24 months group compared to the ≥24 months group (RR 0.59, 95% CI 0.33 to 1.05; participants = 2271; studies = 4; I^2^ = 92%, *p* = 0.07) (Figure 10). However, after removing Peres et al. (2017) during sensitivity analysis, the heterogeneity reduced significantly (I^2^ = 6%, *p* = 0.34). The pooled estimates showed that children breastfed for <24 months had significantly fewer caries compared with children who were breastfed for ≥24 months (RR = 0.49, 95% CI 0.39 to 0.62).

The results of the Hartwig et al. (2019) [43] study are presented narratively due to the method of reporting the outcome. The authors found a statistically significant increase in dental caries in children breastfed for ≥24 months compared to those who were breastfed for less than 6 months or not breastfed (aRR = 8.29; 95% CI 1.82–37.72).

## 4. Discussion

The meta-analyses of the cohort and case-control studies failed to demonstrate a statistically significant difference in dental caries rates between breastfed and non-breastfed groups. However, the systematic review by Cui et al. (2017) [24] found breastfed children had a decreased risk of ECC compared with the children who were never breastfed. Similarly, the systematic review by Avila et al. (2015) [22] found breastfeeding was more protective against ECC than bottle feeding. Similarly, Klaiban et al. (2021) [25] concluded breastfeeding to be protective against ECC in a sufficiently breastfed group compared to a less sufficiently breastfed group. The systematic review by Tham et al. (2015) [28] was inconclusive about whether breastfeeding was associated with ECC. Bagher et al. (2013) [23] indicated exclusive breastfeeding is not associated with caries; however, they postulated that there was a possible protective effect of breastfeeding against ECC. The exclusion of studies with critical bias in the present review reduced heterogeneity, with the sensitivity analysis indicating potential protective effects of breastfeeding against ECC.

Despite breastfeeding being a personal choice, it is profoundly influenced by culture and society [67,68]. Mothers in low-income countries [68] tend to follow the WHO guidelines for breastfeeding, which are exclusive breastfeeding for 6 months and continuing breastfeeding for 2 years or more [18]. Likewise, in countries like Brazil, breastfeeding is culturally expected from mothers, whereas in developed countries like France, it is considered a personal choice. The potential inverse association between breastfeeding and ECC contributes to the broader health benefits of breastfeeding, potentially encouraging increased breastfeeding rates, especially in developed nations where rates are declining [68].

The frequency and duration of breastfeeding may also be influenced by the geographical location of birth, as indicated by studies by Odeniyi et al. (2020) [69] and Pastorelli et al. (2019) [67]. In many countries, the practice of on-demand or frequent breastfeeding [70,71] is prevalent, yet there is often insufficient emphasis on oral hygiene. The frequent exposure to fermentable carbohydrates, such as breast milk, and a lack of effective methods to remove the exposure from the oral cavity are the perfect amalgamation for the development of dental caries. This especially holds true for nocturnal breastfeeding, where there is pooling of breast milk inside the infant’s mouth [27] and decreased salivary flow to wash out the substrate (milk). Thus, exposing the teeth for a prolonged duration to the acid produced by the cariogenic bacteria of the oral cavity results in dental caries. Nocturnal breastfeeding, identified as a risk factor for ECC in this review, has been consistently associated with increased caries risk in preschool children in various studies [50,57,60,61,72] and reviews [28,73].

A study conducted in Cambodia reported that most mothers believed that sucking at night helped children sleep, and more than half of the children who engaged in nocturnal suckling experienced ECC [74]. Similarly, van Palenstein Helderman et al. (2006) [54] observed that all the children who went to sleep with the breast nipple in their mouth experienced ECC. This highlights the need for increased awareness and education regarding the importance of oral hygiene practices, especially in the context of breastfeeding practices, to mitigate the risk of dental caries in preschool children.

Oral hygiene practices should ideally commence from birth, involving the cleaning of gums after each feeding [75]. Special attention to oral hygiene becomes crucial as primary teeth erupt (around 6 months) [76] or when infants start consuming solid food (around 6 months of age). Although cohort studies in this review did not show significant differences in the prevalence of ECC among children in the 4-month or 6-month groups, the meta-analysis of the two case-control studies [60,62] indicated that breastfeeding for less than 6 months is protective against caries. Notably, none of the identified systematic reviews have reported on the association between ECC and breastfeeding for less than or ≥6 months. Nonetheless, Branger et al. (2019) [77], Tham et al. (2015) [28], and Cui et al. (2017) [24] have indicated in their reviews that breastfeeding for < 12 months has a possible protective effect against dental caries. However, the protective association appears to diminish when breastfeeding extends beyond 12 months, and the current review aligns with these findings. Similar results were noticed in this review as well. Similarly, breastfeeding for less than 12 or 18 months may incur a protective effect against ECC, although this association seems to dissipate when the breastfeeding duration reaches 24 months. This outcome concurs with another systematic review [26] reporting that breastfeeding up to 24 months is not associated with a higher risk of ECC. Breast milk is known to inhibit the attachment of some bacteria, such as *Streptococcus mutans* and *Candida albicans* [78], which are known to cause dental caries [79]. However, it promotes the growth of other caries-causing *Streptococcus and Actinomyces* species [78]. Prolonged breastfeeding, therefore, could introduce bacteria that can cause ECC to the sensitive oral microbiome of the infant. There is also a possibility of vertical transmission of the oral pathogens from a mother with an active carious lesion.

Dental caries is a multifactorial disease influenced by host factors, oral bacteria, exposure time, and dietary habits. The escalating prevalence of ECC with age may be attributed to an increase in the number of teeth in the oral cavity [80] and the transition from an exclusive milk diet to a mixed diet incorporating solid foods [81]. Manohar et al. (2021) [82] noted that, beyond the age of two, as children gain autonomy, they tend to opt for unhealthy foods rich in saturated fat and free sugars, contributing to the development of ECC. This underscores the complex interplay of several factors in the aetiology of dental caries in early childhood. Nevertheless, the interpretation of the results of this review warrants caution due to the high heterogeneity observed among the included studies. The heterogeneity was expected due to the lack of homogeneity in the definition of both the exposure and outcome, as mentioned in several systematic reviews [22,23,27,28,29] as well as in some literature reviews [70,77].

## 5. Strengths and Limitations

The strengths of this review are a comprehensive search of five databases without any date or language restrictions and the inclusion of only the cohort and case-control studies, which are considered high-quality studies after experimental studies. Due to the nature of the study, it was determined a priori that RCTs would not be included as breastfeeding should not be randomised as it was deemed unethical by the authors. The critical appraisal was conducted by two reviewers using JBI SUMARI. This review provides evidence from ten meta-analyses showing the association between breastfeeding and ECC.

There were a few limitations while conducting this systematic review. Even though the search was conducted without any language restrictions, due to the limited resources, only the studies in English were included. Due to the exclusion of grey literature and studies in other languages, selection bias cannot be denied. Furthermore, the studies included in this review were mostly conducted in high- or middle-income countries. The search did not identify any studies included in low-income countries, which could result in publication bias. Another possible limitation is the utilisation of unadjusted data in instances where adjusted data were unavailable, which could impact the robustness of the findings. However, the inclusion of adjusted and unadjusted data in the meta-analysis could, in effect, introduce another source of bias.

One of the challenges while conducting this review was the lack of homogeneity of the included studies. Studies used different methods to measure exposure (breastfeeding) and outcomes (dental caries). Valaitis et al. (2000) [29], Peres et al. (2018) [70], and Branger et al. (2019) [77] voiced similar challenges in their reviews. The heterogeneity in the designs of the studies and methods for reporting the outcome measures presumably contributed to the high heterogeneity in several of the meta-analyses. For example, some limitations in the included studies may affect the outcome of the studies and hence influence their interpretation. Most of the challenges were centred upon the definition of the exposures— breastfeeding frequency, duration, and time—and outcomes (dental caries (ECC or S-ECC)). Additionally, the included studies were not clear about exclusive breastfeeding and predominant breastfeeding. Even though most of the studies used WHO criteria to define dental caries, some modifications were also noted. Some did not consider the initial white lesion as caries; some did not consider missing teeth (due to caries); and some only considered frank cavitation. There were inconsistencies regarding how the dental examinations were conducted as well. Some of the studies dried the tooth before examination (showing initial white lesions), while some utilised a visual aid (mouth mirror) only to detect caries (cannot differentiate between stains and caries or caries covered in plaque or food might not be visible), and there were others that used dental probes as well. Some of the dental examinations were conducted in clinical settings, and some were conducted in field settings. All these factors result in an underestimation or overestimation of the findings. Therefore, the results of this review should be interpreted with caution. It was also noted that some of the studies did not differentiate between the genders of the participants. While this lack of information did not affect the result of this review, it could be a possible limitation for anyone who would want to compare the prevalence of ECC between the sexes.

## 6. Recommendations for Future Policy, Practice, and Research

Based on the findings of this systematic review, it is recommended to use standardised definitions regarding duration, frequency, and type of breastfeeding, as well as standardised measurements (without variations) for dental caries assessment, including trained and calibrated dental professionals. Furthermore, clearly defining the minimum intra- and inter-examiner Cohen’s kappa coefficient would help improve the quality of the data collected. A universal and standard method for dental examinations that can be applied in high-income as well as low-income settings would help in maintaining the quality of the studies in different settings.

During this review, it was noted that there was a lack of standard methods for measuring breastfeeding (duration, frequency, and type) in the included studies. Hence, the authors believe following one standard method (without any variations) to diagnose dental caries in future studies would provide a more comprehensive result regarding breastfeeding and ECC. It is also recommended that all the confounders related to breastfeeding and ECC be identified and adjusted to present more valid and reliable results. There is also a need for prospective birth cohort studies in low-income countries to fill the missing gap in the relationship between breastfeeding and ECC.

The findings of this review recommend that policy makers develop a comprehensive oral health promotion programme that includes behavioural modification (like wiping the gums after feeding or offering water after feeding to rinse out milk) to ensure optimum oral health outcomes for preschool children without compromising the nutritional and developmental benefits of breastfeeding. Health professionals such as general practitioners, midwives, and community health nurses should be trained to educate parents on proper oral hygiene from infancy. Since these health professionals take care of the mother and the baby during the antenatal and postnatal stages, oral health awareness can be effectively disseminated among families, particularly the mothers. Dental caries depends on exposure time. Hence, it is recommended to follow good oral hygiene practices as well as behaviour modifications such as minimising on-demand nocturnal feeding and sleeping with a breast nipple in the mouth. Cleaning the teeth with a wet cloth or offering water (to infants > 6 months old) after each feed could also minimise the exposure time. Visiting a dental professional after the child’s first birthday could enhance the parents’ oral health knowledge.

## 7. Conclusions

Even though the meta-analyses failed to demonstrate any statistically significant difference in the risk of ECC between the breastfed and non-breastfed children, they exhibited that breastfeeding for less than 24 months does not appear to increase the risk of ECC; in fact, it may exert a protective effect against ECC. However, it is crucial to note that breastfeeding nocturnally elevates the risk of dental caries in preschool children. Nonetheless, caution needs to be exercised while interpreting the results of this review due to the high heterogeneity. To establish a more comprehensive understanding of the relationship between breastfeeding and ECC, further research is required, employing consistent methods and addressing all the confounders. Until the results of such studies are published, which can provide a more assertive answer regarding the relationship between breastfeeding and ECC, health professionals should follow the recommended guidelines of the WHO and the guidelines of their local health ministry. They should encourage mothers to continue to breastfeed as long as they desire to provide the infants with the benefits of breastmilk, as well as provide oral health education to the mothers that focuses on good oral hygiene practices for the baby and healthy food practices to decrease the risk of ECC.

## Figures and Tables

**Figure 1 nutrients-16-01355-f001:**
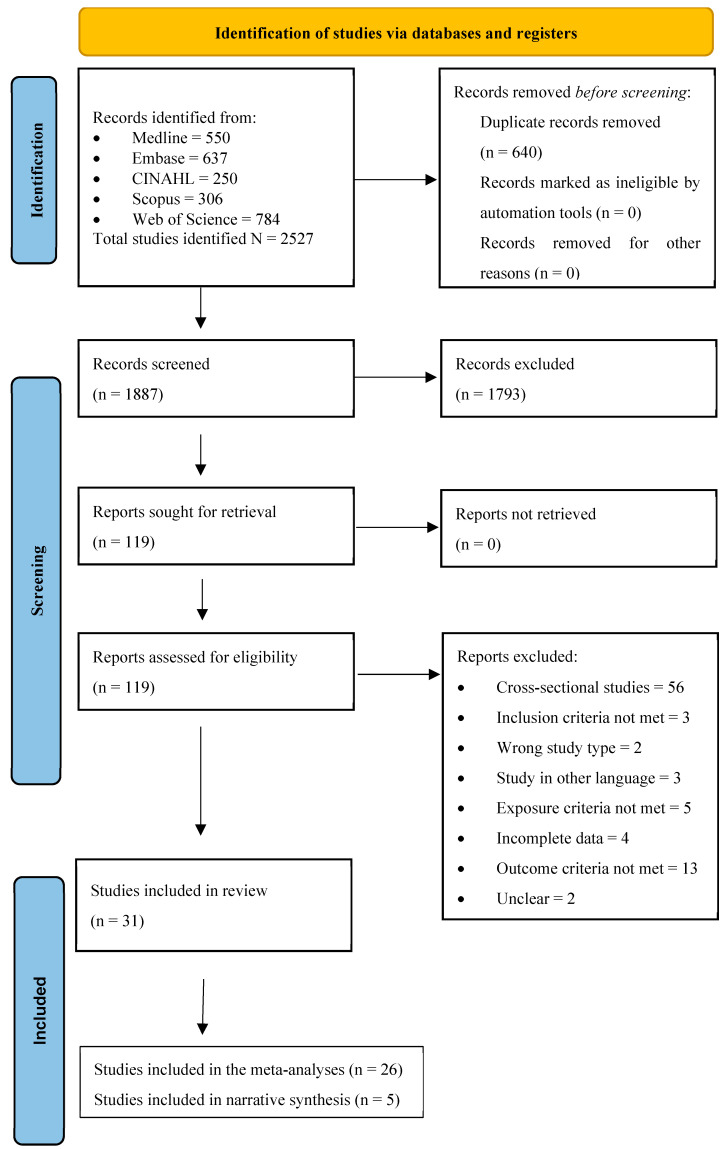
PRISMA flow diagram exhibiting steps in the selection process.

**Figure 2 nutrients-16-01355-f002:**
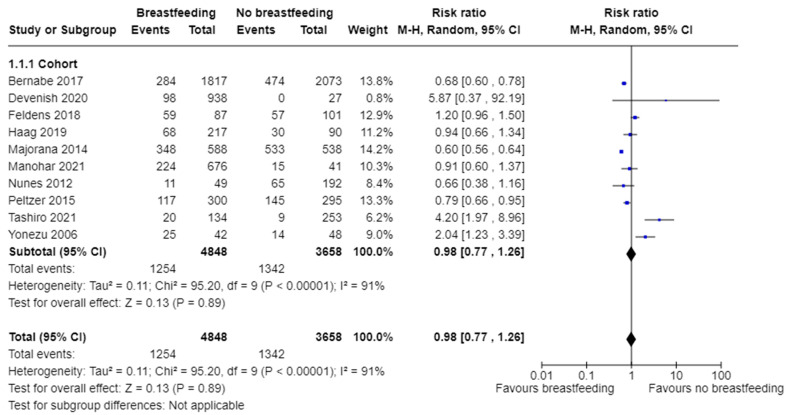
Breastfeeding versus no breastfeeding and the risk of dental caries: cohort studies [37,39,41,42,45,46,49,50,53,56].

**Figure 3 nutrients-16-01355-f003:**
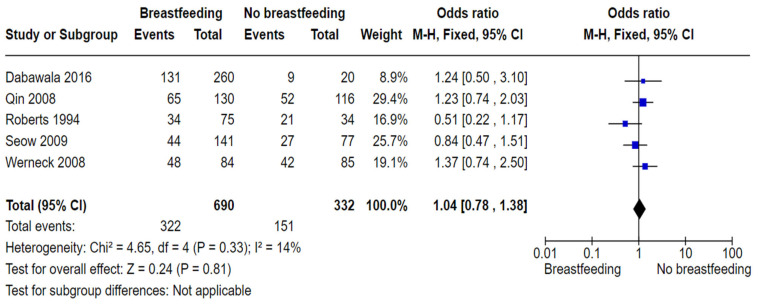
Breastfeeding versus no breastfeeding and the risk of dental caries: case-control studies [58,62,64,65].

**Figure 4 nutrients-16-01355-f004:**
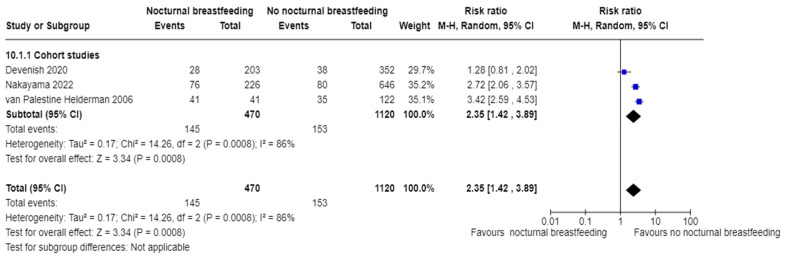
Nocturnal breastfeeding versus no nocturnal breastfeeding and the risk of dental caries [39,47,54].

**Figure 5 nutrients-16-01355-f005:**
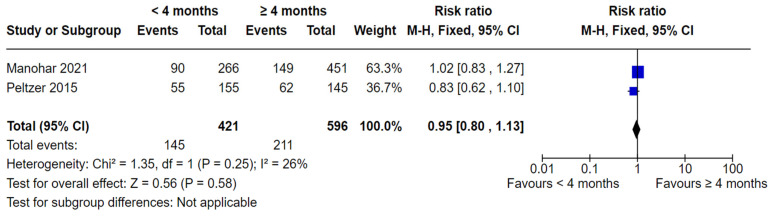
Breastfeeding < 4 months vs. breastfeeding ≥ 4 months and the risk of dental caries: cohort studies [46,50].

**Figure 6 nutrients-16-01355-f006:**
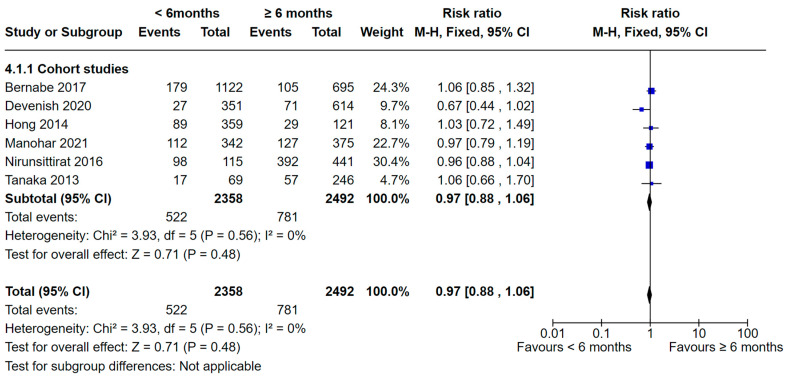
Breastfeeding < 6 months vs. breastfeeding ≥ 6 months and the risk of dental caries: cohort studies [37,39,44,46,48,52].

**Figure 7 nutrients-16-01355-f007:**
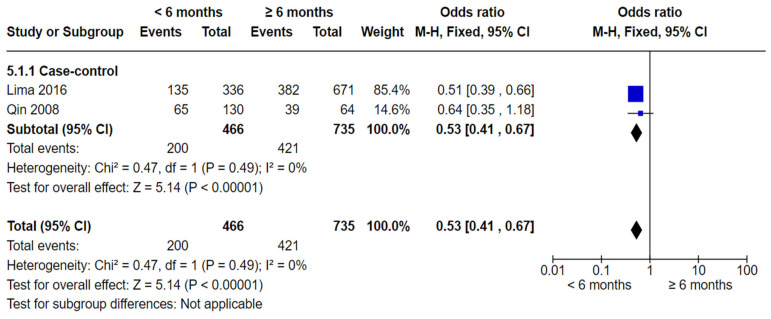
Breastfeeding < 6 months vs. breastfeeding ≥ 6 months and the risk of dental caries: case-control studies [60,62].

**Figure 8 nutrients-16-01355-f008:**
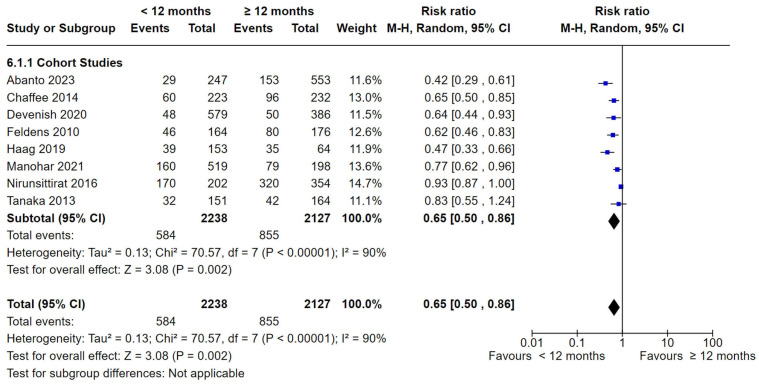
Breastfeeding < 12 months vs. breastfeeding ≥ 12 months and the risk of dental caries: cohort studies [35,38,39,40,42,46,48,52].

**Figure 9 nutrients-16-01355-f009:**
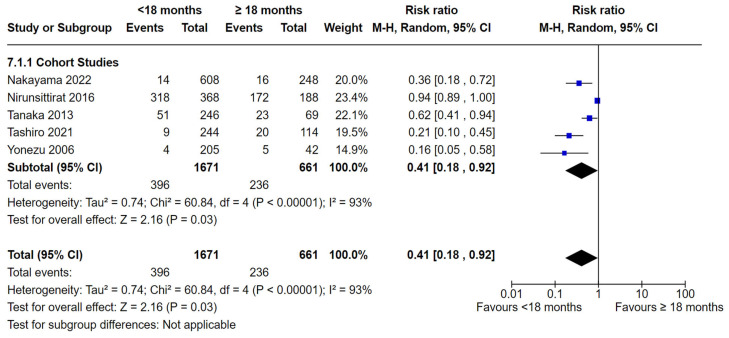
Breastfeeding < 18 months vs. breastfeeding ≥ 18 months and the risk of dental caries: cohort studies [47,48,52,53,56].

**Figure 10 nutrients-16-01355-f010:**
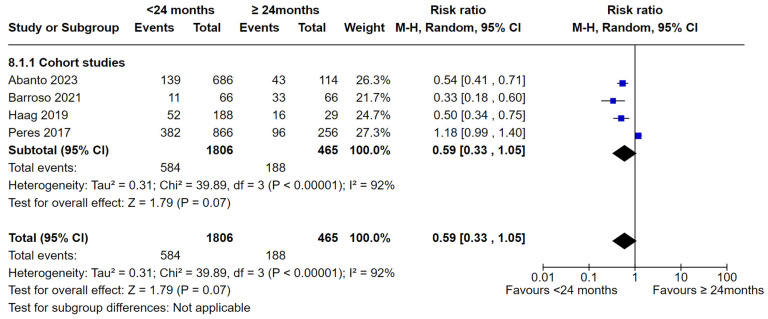
Breastfeeding < 24 months vs. breastfeeding ≥ 24 months and the risk of dental caries: cohort studies [35,36,42,51].

**Table 1 nutrients-16-01355-t001:** List of inclusion and exclusion criteria according to P(I/E)CO criteria.

	Inclusion Criteria	Exclusion Criteria
Population	Children under 6 years of age (preschool children) Any gender, race, geographical location, or socioeconomic status Without any systemic disease or disability	Children who are 6 years or older Children with any systemic disease or disability
Exposure	Preschool children (<6 years old) who are breastfed (exclusive or partial)	
Comparator	Preschool children (<6 years old) who are not breastfed	
Outcome	Studies reporting the prevalence of dental caries (as reported by a qualified dental practitioner or trained and calibrated non-dental personnel) Studies reporting dental caries only on primary teeth in preschool children First dental examination before the age of six	Studies not reporting dental caries Self-reported or unvalidated dental caries measures Studies reporting dental caries in permanent teeth First dental examination at 6 years or older
Study design	Cohort studies Case-control studies Cohort and case-control studies nested in other study designs	Randomised controlled trials (RCTs) Quasi-experimental studies Cross-sectional studies Single case reports Grey literature Systematic reviews, literature reviews, umbrella reviews, and scoping reviews

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
