# Peer review of "Association of Breastfeeding and Early Childhood Caries: A Systematic Review and Meta-Analysis"

_nutrients, 2024, doi:10.3390/nu16091355_

Round 1

Reviewer 1 Report

Comments and Suggestions for Authors

We read with great interest the manuscript with title “Association of breastfeeding and early childhood caries: A systematic review and meta-analysis” aiming to synthesises the evidence on the association between breastfeeding and ECC.

1. Introduction: the length of the text of the Introduction should be halved, and the parts you remove can be moved into the discussion

2. Throughout the text, avoid the use of personal pronouns in the first-person plural, such as 'we' and the corresponding adjective 'our'.

3. In the discussion you are asked for a paragraph to be added on the possible relationship between prolonged breastfeeding and the creation of the oral microbiota community in the child, it could be hypothesised that prolonged breastfeeding may be related to a more sensitive infant oral microbiota for early childhood caries. Indeed, the diet during early childhood shapes the early oral microbiota formation, and perhaps breastfeeding favours some species over others, depleting the microbiota in its complexity. Just as not having data on the mother's oral health cannot exclude the possibility that if the mother has caries in progress, she vertically transmits to her child in early childhood a microbiota more susceptible to caries. In any case, it is a very fascinating scientific question that broadens the horizons of the discussion. In this regard, it is worth mentioning this article (10.3390/ijerph18115569), which is one of the few that has posed these questions.

4. Results: I would like the authors to add the characteristics of the included studies, giving the reader more information about the period in which the included studies were conducted, the countries and continents, and the setting (hospital/university, etc…). In addition, I would like to ask the authors if they found any studies that distinguished between boys and girls and if there are any gender-based data.

Author Response

Thank you very much for taking the time to review this manuscript. Please find the detailed responses in the attachment and the corresponding revisions/corrections in track changes in the re-submitted files

We read with great interest the manuscript with title “Association of breastfeeding and early childhood caries: A systematic review and meta-analysis” aiming to synthesises the evidence on the association between breastfeeding and ECC.

Comment 1. Introduction: the length of the text of the Introduction should be halved, Cand the parts you remove can be moved into the discussion

Response 1: We have shortened the introduction. Please see page 2, lines 57- 66, 88- 94.

Comment 2: Throughout the text, avoid the use of personal pronouns in the first-person plural, such as 'we' and the corresponding adjective 'our'.

Response 2: Thank you for pointing it out. Corrections were made accordingly.

Comment 3: In the discussion you are asked for a paragraph to be added on the possible relationship between prolonged breastfeeding and the creation of the oral microbiota community in the child, it could be hypothesised that prolonged breastfeeding may be related to a more sensitive infant oral microbiota for early childhood caries. Indeed, the diet during early childhood shapes the early oral microbiota formation, and perhaps breastfeeding favours some species over others, depleting the microbiota in its complexity. Just as not having data on the mother's oral health cannot exclude the possibility that if the mother has caries in progress, she vertically transmits to her child in early childhood a microbiota more susceptible to caries. In any case, it is a very fascinating scientific question that broadens the horizons of the discussion. In this regard, it is worth mentioning this article (10.3390/ijerph18115569), which is one of the few that has posed these questions.

Response 3: We have now added a paragraph the possible vertical transmission from mother to her child. See page 16 lines 574- 580.

Comment 4. Results: I would like the authors to add the characteristics of the included studies, giving the reader more information about the period in which the included studies were conducted, the countries and continents, and the setting (hospital/university, etc…). In addition, I would like to ask the authors if they found any studies that distinguished between boys and girls and if there are any gender-based data.

Response 4: We have now added the countries where the studies were carried out. See page 7, lines 273- 282, 295 to 308. Some of the studies did not indicate where the study setting was, and almost half of the studies did not distinguish between the gender of the participants. We have noted this as a limitation in the review. Please see page 17 lines 633- 636.

Reviewer 2 Report

Comments and Suggestions for Authors

The meta-analyses of the cohort and case-control studies failed to demonstrate a statistically significant difference in dental caries rates between breastfed and non-breastfed groups

The topic is very interesting and ambitious I should add the PICO question as it would better focus the topic of study 

The paper is  well written and follows the rules of a meta-analysis, however, it should be considered:

Should define variables for comparability between studies.

Caution should be taken in interpreting results due to high heterogeneity. 

 It is recommended to give guidance and include specific recommendations for clinicians to follow.

Give a thorough exploration of possible biases.

 The inclusion of studies in other languages

Tables 2 and 3 should be improved. 

Author Response

Thank you very much for taking the time to review this manuscript. Please find the detailed responses in the attachment and the corresponding revisions/corrections in track changes in the re-submitted files

Comment 1: The meta-analyses of the cohort and case-control studies failed to demonstrate a statistically significant difference in dental caries rates between breastfed and non-breastfed groups.

Response 1: We did not find an association and have reported in results and discussion. We have added this to conclusion as well (Page 18 lines 670- 671)

Comment 2: The topic is very interesting and ambitious I should add the PICO question as it would better focus the topic of study 

We have added the research questions as well as structured the inclusion criteria as P(I/E)CO criteria. Please see page 3 lines 122- 127, 140; page 4, lines 148-149.

Comment 3: The paper is well written and follows the rules of a meta-analysis, however, it should be considered:

Comment 3a: Should define variables for comparability between studies.

Response 3a: Thank you for the comment. We have now identified different variables in the method section. Please see page 6 lines 224- 228.

Comment 3b: Caution should be taken in interpreting results due to high heterogeneity. 

Response 3b: Thank you for your comment. We already indicated this in page 16, line 589-590. We have added to the conclusion as well (Page 18 lines 674- 675).

Comment 3c: It is recommended to give guidance and include specific recommendations for clinicians to follow.

Recommendations for dental professionals are indicated in page 17, lines 639- 641. Recommendations for other health professionals added on page 18 lines 658-662.

Comment 3d: Give a thorough exploration of possible biases.

The biases have been reported on the 2nd paragraph on page 17. We are not sure if any other biases were noted.

Comment 3e: The inclusion of studies in other languages

The studies in other languages were not included due to limited resources, as indicated in page 17, lines 603-605. We have noted this as a limitation.

Comment 3f: Tables 2 and 3 should be improved. 

They have been moved to a supplementary file (Appendices; Tables C3 & D4).
